# The Coordinated Changes in Platelet Glycan Patterns with Blood Serotonin and Exosomes

**DOI:** 10.3390/ijms252211940

**Published:** 2024-11-06

**Authors:** Fusun Kilic

**Affiliations:** Retired Professor of Biochemistry and Molecular Cellular Biology; fusunkilic@yahoo.com

**Keywords:** 5-HT, platelet, SERT, exosome, glycans, platelet physiology, 5-HT signaling

## Abstract

The structures of glycans, specifically their terminal positions, play an important role as ligands for receptors in regulating the adhesion ability of platelets. Recent advances in our understanding of free/unbound serotonin (5-HT) in blood plasma at supraphysiological levels implicate it as one of the most profound influencers in remodeling the platelet’s surface *N*-glycans. Proteomic analysis of the membrane vesicles identified enzymes, specifically glycosyltransferases, only on the surface of the platelets isolated from the supraphysiological level of 5-HT-containing blood plasma. However, these enzymes can only be effective on the cell surface under certain biological conditions, such as the level of their substrates, temperature, and pH of the environment. We hypothesize that exosomes released from various cells coordinate the required criteria for the enzymatic reaction on the platelet surface. The elevated plasma 5-HT level also accelerates the release of exosomes from various cells, as reported. This review summarizes the findings from a wide range of literature and proposes mechanisms to coordinate the exosomes and plasma 5-HT in remodeling the structures of *N*-glycans to make platelets more prone to aggregation.

## 1. Serotonin (5-HT) in Blood Plasma

Serotonin, 5-HT (5-Hydroxytryptamine), is an intermediate product of tryptophan metabolism, and is primarily located in the enterochromaffin cells of the intestine [1,2,3,4], serotonergic neurons of the brain [5,6], and platelets of the blood [7]. Cigarette smoking [8,9,10] and episodic incidents [11,12,13,14,15], such as stress, anxiety, or antihypertensive drugs, produce an overactive sympathetic nervous system (SNS)*,* which stimulates the secretion of 5-HT from the enterochromaffin cells into the blood to supraphysiological levels [16]. The free/unbound 5-HT is taken from the blood plasma by platelets expressing serotonin transporters (SERTs) [7]. SERT is the major mechanism in regulating the plasma 5-HT level by a saturable reuptake mechanism [6,7]. Once in the platelet cytoplasm, 5-HT is sequestered by the vesicular monoamine transporter type 2 (VMAT2) into intracellular dense granules [17,18,19,20,21,22,23], but prior to sequestration, the 5-HT molecules can activate intracellular signaling pathways linked to platelet activation and aggregation [24,25,26,27,28,29]. Notably, plasma 5-HT is in the low nanomolar range, but the dense granules of resting platelets store millimolar concentrations of 5-HT [17,18,19,20,21,22,23]. Thus, platelets are designed to tightly control the plasma 5-HT concentration. Therefore, the regulation of the SERTs’ activity constitutes an important mechanism in controlling the action of the 5-HT signaling; otherwise, while the plasma 5-HT reaches a supraphysiological level, the 5-HT signal becomes constitutively active [30,31,32].

Plasma 5-HT levels play a critical role in almost all cardiovascular disorders, including hypertension [11,12,13,14,15,16], coronary artery disease [32,33,34,35], atherothrombosis [36], and myocardial infarction [37,38,39]. The free/unbound level of 5-HT in the neuronal system is associated with various neuropsychiatric diseases and disorders [6,40,41]. In both the cardiovascular and nervous systems, SERT plays a key role in regulating the extracellular free 5-HT levels [42,43]. 

In a similar manner, the 5-HT uptake rates of cells are proportional to the number of SERT molecules assembled in the plasma membrane, although this is not the only factor. The posttranslational modifications, folding, and correct assembly into the plasma membrane also play a key role in the 5-HT uptake rates of SERT [29,44,45,46,47,48]. But intriguingly, the surface expression of SERT molecules on neurons and glial cells is regulated by the concentration of extracellular 5-HT [49]. The number of SERT molecules in the plasma membrane of these cells is changed relative to the concentration of the extracellular 5-HT [44,45,46,47]. Thereby the high levels of 5-HT in extracellular locations as regulatory signaling molecule, limit the synaptic availability through altering the trafficking dynamics of SERT via 5-HT receptors and second messengers [26,40,41]. 

In episodic hypertension, the plasma 5-HT levels elevate to 33% of normotensive levels [13,50]; following myocardial infarction, which was associated with coronary artery disease and cardiac events, a 10–fold elevation in plasma 5-HT levels was reported by Vikens et al. [35]. In the coronary sinus of patients, following angioplasty, 5-HT level was reported as 16-fold increased [39]. However, in systemic plasma, the measurement of 5-HT concentration appeared normal, suggesting that the elevation in 5-HT concentration could only be local, which may not be routinely detected as a risk factor. Thus, a physiological in vivo interplay between circulating 5-HT and platelet function may be a predictor of coronary artery disease. Additionally, atherothrombosis, cerebrovascular ischemia, and myocardial infarction have been linked to elevated 5-HT levels [11,12,13,14,15,16,17,18,19]. Indeed, tryptophan hydroxylase 1 (TPH1) knockout (KO) mice lacking 5-HT have a mild bleeding abnormality [30,31,51]. Notably, patients diagnosed with cardiovascular disorders also may show a blunted release of endothelium-derived prostacyclin and nitric oxide [19]. The deficit of these anticoagulant molecules was proposed as permitting the amplification of the pro-coagulant actions of 5-HT [52]. Therefore, the major questions concerning the plasma 5-HT level are how its diverse functions are mediated in different cardiovascular diseases at different stages of their development, as well as seeking the causes of the downregulation of the 5-HT uptake rates of SERT. 

## 2. 5-HT Signaling and Platelets Physiology

In the process of dissecting the roles of 5-HT signaling on cellular physiology, the first step should be learning how the recycling of SERT in platelets correlates with the levels of free/unbound 5-HT in plasma/extracellular compartments. This might shed some light on other proteins and glycans trafficking between intracellular compartments and the plasma membrane. 

In vitro studies have demonstrated that the pretreatment of platelets with 5-HT increases the density of SERT molecules in the plasma membrane [11]. In episodic hypertension, the platelet 5-HT content decreased by 33% and the plasma 5-HT content increased by 33%, while there was a decrease in Vmax but with a similar Km, primarily due to a decrease in the number of SERT molecules on the platelet surface (B_max_) (Figure 2 of [11]). Therefore, in episodic hypertension, the unbound 5-HT at supraphysiological level establishes a continuous 5-HT-signaling that downregulates the 5-HT uptake ability of the platelets via decreasing the density of SERT molecules on the plasma membrane. The signaling of 5-HT plays an important role in platelets’ membrane trafficking pathways, i.e., internalization, recycling and the subcellular redistribution of membrane proteins and glycans [13,25,26,44,45,46,47,48]. While the density of SERTs on the platelet surface decreased, no changes were found in the whole cell expression of SERT in platelets under hypertensive conditions [13]. 

Like the other membrane proteins, SERT is packed in a secretory vesicle and either stays in the cytoplasm or moves to the plasma membrane, depending on the plasma/extracellular 5-HT level. The number of available transporters on the plasma membrane defines platelets’ 5-HT uptake rates. Platelets are an excellent model to study the relationship between plasma 5-HT levels and 5-HT uptake rates via investigating the membrane trafficking of SERT. This is because, in platelets, the biosynthesis of proteins is minimal; it is much lower than the rate of their degradation. Studies have reported that the synthesis and modification of proteins still occur in platelet cytoplasm at a lower rate [53]. However, the cellular steps and factors involved in the biosynthesis, posttranslational modifications, and correct folding of SERT occur in megakaryocytes [54]. When the platelets are formed and separated from the megakaryocyte, SERT in small vesicles is translocated from the cytoplasm to the plasma membrane. We compared the global gene expressions in megakaryocytes of TPH1-KO vs. wild-type mice; the expression of the SERT gene in TPH1-KO mice megakaryocytes was lower than the levels in wild-type mice megakaryocytes [54]. These studies weaken the hypothesis of the biosynthesis of SERT in platelets.

The capacity for 5-HT uptake is related to its density on the plasma membrane of the proteins, which play important roles in their functions that regulate the translocation of SERT to/from the plasma membrane [24,25,26,28,29,45,48]. Therefore, the density of SERT molecules on the platelet surface is the major factor in their 5-HT uptake rates. 

Once 5-HT is taken into the platelets’ cytoplasm from the blood plasma, SERT is removed from the plasma membrane through endocytosis; first, it is sequestered in a small vesicle and then recycled in the cytoplasm [45,48]. Deleting SERT gene in mouse model (SERT-KO) [43] helped to elucidate the relationship between platelet SERT expression, circulating 5-HT levels in plasma, and the contribution of these influences to platelet physiology [45,48]. For example, SERT-KO platelets were completely depleted of 5-HT, which confirmed the role of platelet’s SERT in dense granules packed 5-HT which is taken in from the blood plasma [54]. 

Studies on the platelets of mice lacking the gene for TPH1, which is the rate-limiting enzyme in the synthesis of 5-HT in peripheral cells, demonstrated a requirement for intracellular 5-HT together with Ca^2+^ for the release of α-granules during platelet activation [30,31,46]. These studies on isolated platelets indicated that 5-HT-mediated signaling accelerated the exocytosis of α-granules, which secrete their ingredients, the procoagulant molecules, into plasma [30,31,46]. These molecules may include fibrinolytic regulators, growth factors, chemokines, immunologic modulators, and adhesion molecules, such as P-selectin, von Willebrand factor, thrombospondin, fibrinogen, and fibronectin [18]. Therefore, these findings confirm an important regulatory role for elevated extracellular 5-HT in the membrane trafficking of intracellular components, including granules. 

However, determining the precise factors that initially increase the plasma 5-HT levels in cardiovascular disease patients is difficult to pinpoint. Clinical and preclinical studies report that episodic incidents produce overactive SNS, stimulating the secretion of 5-HT into the blood at supraphysiological levels [8,9,10]. Regardless of the reason for the elevation in plasma 5-HT levels, there is a resulting change in platelet physiology, specifically in the intracellular membrane trafficking pathways [9,13,25,29,45]. 

While 5-HT is well studied as a neurotransmitter, it is not surprising that most of the 5-HT receptors are expressed outside of CNS [24]. In muscle cells, it was shown that 5-HT-mediated signaling activates p21 activating kinase (PAK), which in turn phosphorylates vimentin on the serine residue at position 56 [55,56]. Vimentin, an intermediate filament, is found in platelets and plays a role in their adhesive feature [57,58]. Following phosphorylation, the curved filamentous structure of vimentin undergoes reorganization and straighten [55,56,57,58,59]. In its straightened form, phosphovimentin can bind to SERT with a higher affinity, which accelerates the internalization of SERT from the plasma membrane [29,45]. In general, the translocation of proteins to/from the plasma membrane is mediated by their partner proteins that facilitate the movements of the membrane-located proteins. Rab4 and PAK-phosphorylated vimentin were reported as the two molecular pathways that alter SERT trafficking in response to elevated plasma 5-HT [29,45,48]. 

These in vitro findings were supported by the findings from 5-HT-infused mice [46]. When the aggregation rates of platelets from saline- and 5HT-infused mice were compared, the data showed that 50% of the platelets of the saline-infused mice were aggregated at the end of 4 min of stimulation with fibrillar collagen, and that 90% of the platelets from 5HT-infused mice were aggregated in the same period [46]. These data support the findings from the in vitro studies on elevated plasma 5-HT levels downregulate the surface density of SERT, resulting in platelet depletion of 5-HT [13] and amplified aggregation responses [46]. Thus, the downregulation of SERT on the surface of platelets contributes an additive influence on platelet aggregation as “platelets to plasma” shift of 5-HT. This effect would be one of many changes in platelets, which collectively would influence platelet function. The proposed mechanism shown in Figure 1 is quite intriguing and illustrates how the levels of 5-HT in blood plasma control the density of SERT molecules on the platelet surface. 

Traditionally, it has been described that SERT on the plasma membrane regulates the level of plasma 5-HT via a saturable uptake mechanism; however, these studies provide evidence that 5-HT at supraphysiological levels promotes its extracellular concentration more by reducing the density of SERT molecules on the plasma membrane via changing the characteristics of SERT partners in the membrane trafficking pathway.

The elevation of plasma 5-HT by osmotic minipump in mice in vivo results in a loss of SERT on the platelet plasma membrane by activating the Rab4-GTP and phosphovimentin pathways [25,46]. This abnormality is associated with increased platelet aggregation assessed by aggregometer studies in vitro. 

Finally, the excess 5-HT enhances platelet aggregation induced by other endogenous substances. However, even at the highest levels of plasma 5-HT, there are always a few SERT molecules on the platelet membrane that continue to clear plasma 5HT, but at a lower rate (V_max_), most probably until the plasma 5-HT levels return to the physiological level and SERT expression on the cell surface is restored [13]. Thus, a highly dynamic relationship appears to exist between plasma 5-HT levels and SERT trafficking that may be capable of influencing platelet function. 

In summary, 5-HT-mediated signal was shown to modify platelet physiology through acting on membrane trafficking, including the downregulation of the density of SERT molecules on the surface, in three study models: the in vitro (5-HT-treated platelets in culture dish), clinical (episodic hypertension) [13], and 5-HT-infused mice model [46]. These findings correlated with the platelet’s 5-HT uptake rates through the downregulating of SERT, which results in a “platelet to plasma shift”. The presence of SERT on the platelet surface and 5-HT in the plasma at an elevated level exhibit a biphasic relationship (Figure 2). This biphasic relationship could be the reason for the receptor-mediated endothelial relaxation ability of 5-HT, as reported by Watts’ group in several nicely designed studies [60]. 

## 3. Glycans Regulates the Adhesion of Platelets 

To gain a better understanding of the impact of plasma 5-HT on the adhesive features of platelets, mice implanted with osmotic mini-pumps that infused 5-HT for 24 h had their platelets isolated from blood samples [46]. The proteomic, mass spectrometry (MS) analysis of the membrane vesicles showed an elevation in the *N*-glycolyl-neuraminic acid (Neu5Gc) containing *N*-glycans on platelets isolated from 5-H-infused mice blood samples [61]. Neu5Gc is formed from *N*-glycolyl-neuraminic acid (Neu5Gc) via the catalytic action of Cytidine monophosphate-N-acetylneuraminate hydroxylase (CMAH). The predominance of Neu5Gc on the platelets of the 5-HT-infused mice suggests the upregulation of CMAH via 5-HT-mediated signaling through the platelet-located 5-HT receptor activity. Further analysis of 5-HT signaling in 5-HT-infused SERT-KO mice showed a 2-fold higher Neu5Gc population on these platelets. Specifically, in 5-HT-infused, SERT-KO mice, the plasma 5-HT should be at the highest, supraphysiological level since the platelets do not express SERT, and 5-HT stays in plasma. The direct correlation between the plasma 5-HT level and the level of Neu5Gc on the platelet surface suggests that 5-HT-mediated signaling may play a role in promoting the catalytic function—not the expression—of CMAH. The data from the study are consistent with fluorescence-activated cell sorting analysis of platelets stained with Neu5Gc antibodies [61].

The supraphysiological level 5-HT dependent remodeling of the platelet *N*-glycan content opens a new era in platelets physiology [9,61]. 5-HT-infused mice, a preclinical study model, provided exceptional insight into plasma 5-HT level-associated platelet aggregation [46]. However, clinical studies reported accentuated elevation in platelets aggregation rates in SNS-mediated episodic hypertension [8,9,10,11,12,13,14,15,50]. 

The plasma 5-HT level is elevated in various conditions, including hypertension [13] and thrombosis [46]. However, in the absence of cardiovascular disease, the in vivo administration of 5-HT does not increase systolic blood pressure [47], suggesting that the elevation in plasma 5-HT levels could be a consequence rather than a cause of some forms of hypertension [50]. The impact of plasma 5-HT on hypertension can vary with its acute or chronic elevation and the location in the circulatory system.

Episodes of stress, anxiety, smoking, or medication-originated conditions activate the SNS, which causes the elevation of unbound/free 5-HT levels in the blood plasma. Clinical findings reported that under episodic conditions the platelet aggregation rates are accelerated, resulting in a hypercoagulable state [13,46,47,48]. One of these episodic conditions is cigarette smoking, which produces central nervous system-mediated activation of the SNS and stimulates the secretion of 5-HT to supraphysiological levels in the blood [9]. The enhanced levels of 5-HT in smokers’ blood are associated with increases in G protein-coupled receptors signaling and the serotonylation of small GTPases [9,25,27,28,29,30,31,44,45,46,48], which in turn lead to the remodeling of cytoskeletal elements to enhance granule secretion and promote the unique expression of sialylated *N*-glycan structures on smokers’ platelets [9,25]. A proteomic analysis identified the enzymes involved in glycan biosynthesis/modification only on smokers’ platelet membranes [9]. Furthermore, enzymatic removal of the *N*-glycan from the platelet surface counteracts smoking-mediated platelet aggregation, emphasizing the involvement of *N*-glycans in the adhesiveness of the platelet [9]. 

Liquid chromatography–mass spectrometry (LC/MS) analysis of proteins extracted from plasma membrane vesicles isolated from smokers’ and non-smokers’ platelets identified several proteins and enzymes involved in glycan biosynthesis/modification only on the smokers’ platelet membranes [9]. Table 1 presents some of these. Five of the six proteins found to be most highly elevated in smokers’ platelets were proteins with well-defined roles in the reorganization of the cytoskeleton, the sixth of which is a modulator of Gi signaling, the G protein. 

MALDI-MS analysis of the surface glycans eluted from platelet plasma membrane vesicles revealed differences in the types of N-glycans from the two populations [9]. Nonsmokers’ platelets had twice as many high mannose-type structures as smokers ‘platelets (~41% vs. ~19%). On the other hand, smokers’ platelets had more total sialylated N-glycan than the nonsmokers’ platelets (~74% vs. ~56%). Among the sialylated glycans, total monosialylated and disialylated species were higher for smokers than for nonsmokers (monosialylated, 19% vs. 8%; disialylated, 47% vs. 36%). The relative abundance of N-glycan was 30% higher in smokers’ platelets than in nonsmokers ’platelets [9].

Interestingly, two glycosylation enzymes, UDP-glucuronate decarboxylase (UXS) and mannosyltransferase (ALG11), were found specifically on the surface of platelets isolated from smokers’ blood samples. Furthermore, removing the *N*-glycans from the surfaces of smokers’ platelets counteracted the smoking-mediated aggregation. Currently, the role of UXS/ALG11 is not known, but they may play a role in the smoking-induced remodeling of the platelet surface *N*-glycan. These glycosylation enzymes in small vesicles are translocated to the platelets’ surface in 5-HT signaling-induced trafficking, which are normally located in intracellular compartments; therefore, they should contribute adhesive properties to glycoproteins on the platelet surface after smoking. While this role is not yet known, it should be studied, because it is important to explore whether the cell-surface high-mannose glycans (present on nonsmokers’ platelets) protect platelets from aggregation. 

The pathway of N-linked protein glycosylation in eukaryotes is highly conserved and starts with the assembly of Glc3Man9GlcNAc2-PP-Dol, the common core oligosaccharide donor, the glycan moiety of which is subsequently transferred by the oligosaccharyltransferase complex onto selected Asn-XSer/Thr acceptor sites of the nascent polypeptide chain. ALG11 is involved in the last steps of the synthesis of Man5GlcNAc(2)-PP-dolichol core oligosaccharide on the cytoplasmic face of the plasma membrane; it catalyzes the addition of the fourth and fifth mannose residues to the dolichol-linked oligosaccharide chain. The enzymatic activity of solubilized ALG11 (in platelets or isolated plasma membrane vesicles) should be determined.

As reported earlier, cigarette smoking stimulates the release of 5-HT to a supraphysiological level; at that level, 5-HT accelerates the dynamics of membrane trafficking in platelets in a G-protein-coupled receptor signaling pathway [9,25,27,28,29,30,31,44,45,46,48]. Despite the intracellular diversity of glycosyltransferases, only UXS and ALG11 were identified on the surface of the platelets isolated from the cigarette smokers’ blood samples. However, the other enzymes that are normally located in platelets but move to the cell surface via 5-HT signaling trafficking remain undefined. Furthermore, it needs to be investigated whether these enzymes are functionally active in the presence of exosomes. 

Enzymatic reactions require certain conditions, such as temperature, substrate vs. enzyme concentrations, pH, and time. In the absence of these conditions, enzymes are just proteins without catalytic properties. On the platelet surface, UXS, ALG11, and the others Need a certain criterion in order to be active. The studies with the cigarette smokers’ platelet plasma membrane were a molecular model to emphasize the 5-HT signaling-related changes on the platelet surface. Cigarette smoking elevated plasma 5-HT to supraphysiological level accelerates the dynamics of membrane trafficking in platelets through G-protein coupled receptor signaling pathway [9,25]. These cellular processes identified UXS and ALG11 only on the surface of the cigarette smokers’ platelets. However, in different diseases and disorders, the platelets’ surfaces should have other enzymes and glycans. These areas should be investigated in detail.

Only a few of the glycosyltransferases are located on the cell surfaces. This then raises the possibility that platelet surface glycosyltransferases modulate the platelet surface glycan structures and functions. The following section will explain how the glycosyltransferases produce glycans on the surface of the platelets; and if exosomes maintain a foundation on platelet surface to support their enzymatic activities. 

## 4. Exosomes 

Exosomes are nanovesicles released from viable cells, either constitutively or upon the activation of cell secretion [62]. These bioactive vesicles reflect the physiological state of the originating cells, thus appearing as a good source for monitoring the development of diseases or medical treatment in real time. Therefore, the exosomes from a patient’s blood samples could be collected at different time points and analyzed directly using omics approaches. For example, placenta-derived exosomes could be a good source for monitoring placental development in real time. Placenta, a unique vascular organ, separates the maternal and fetal circulations and plays a central metabolic role in pregnancy. The healthy development of placenta reflects its functional level, which is directly related to the healthy growth of fetuses. Analyzing the placenta-derived exosomes periodically should present an adequate “real-time” method for monitoring the development and functionality of the placenta; perhaps this lends credence to the Barker hypothesis on the antenatal basis of a number of chronic adult diseases [63]. 

Another example of the role of exosomes is their contributions to platelet aggregation. The MS analysis of the contents of cigarette smokers’ and nonsmokers’ exosomes showed differences in their nucleotide sugar contents [64,65,66]. Interestingly, in our preliminary studies, we observed that incubating nonsmokers’ platelets with smokers’ exosomes elevated the platelet aggregation rate. The nucleotide sugars themselves are critical precursors to decorating the glycoprotein on the cell surface. Such alterations may favor a glycan-rich surface, specifically in xylose or mannose residues on platelets [9]. The summary of these findings shows that cigarette smoking triggers changes in the pool of nucleotide sugars in exosomes in enhancing platelet activation [9,25]. 

Exosomes are formed by budding in intraluminal vesicles to form multivesicular bodies. Some multivesicular bodies will fuse with the plasma membrane and release their intraluminal vesicles or exosomes into the extracellular space. Tissue-derived exosomes, bioactive vesicles, should reflect the physiological state of the originating cells tissues that they derived from, which are a good source in monitoring tissue development in real time. Due to their important role in physiology and intercellular communication, as well as their potential for being a resource for the identification of biomarkers, there is increasing interest in studying exosomes. Recent reviews have emphasized the largely unexploited potential for studying exosomes derived from various tissues [67,68,69] and highlighted their important potential role in processes such as endothelial cell migration [69] and immune modulation of pregnancy success [70]. Proteomics techniques have been applied to exosomes derived from cultured trophoblast cells [71,72,73] and umbilical cord blood [74]. A study of serum-derived microparticles suggested that proteomics methods could provide novel proteomic biomarkers for spontaneous preterm births, although a correlation with placental microparticle markers was not attempted [75]. One recent study demonstrated that hypoxia regulates the response of trophoblast-derived exosomes to hyperglycemia and suggested that there was a unique placental exosome profile in plasma from patients with gestational diabetes mellitus (GDM) [76], but comprehensive protein profiling was not attempted. That study suggests that the study of exosomes from GDM patients is unexplored and has significant potential for generating critical new understandings of GDM. Finally, another study on that subject has demonstrated that placenta-derived exosomes continuously increase in maternal circulation over the first trimester of pregnancy, based on the increased detection of the placenta-specific marker, placental alkaline phosphatase (PLAP), in exosomes from maternal peripheral blood [77]. That study was the first to use proteomic techniques to look at the protein cargo of exosomes in peripheral blood with a probable placental origin, and more than 340 proteins were detected. In addition, the study of protein profiles, specifically glycosylation, is emerging as a posttranslational modification (PTM) that is particularly important to the proposed work. Recent studies have indicated the N-Linked glycans have an important role for protein cargo recruitment into exosomes [78], i.e., ovarian carcinoma cells release exosomes with specific glycosignatures have been proposed as a source of biomarkers [79]. Placental tissue should be studied via proteomics techniques, including 2-dimensional gel electrophoresis techniques [80,81,82,83,84,85] and shotgun mass spectrometry-based proteomics [86,87,88]. Some have tried to analyze serum or plasma as a source of placental biomarkers, especially for pre-eclampsia [89,90,91], but these efforts have been hampered by interference from high abundance proteins found in serum [92]. A recent review of mass spectrometry-based proteomics for studying pre-eclampsia and preterm birth included many studies related to the role of the placenta, especially in pre-eclampsia [92]. The authors noted the relatively poor linearity and quantification accuracy of proteomic methods employing isotopic labeling, such as the iTRAQ and TMT, which resulted in “…only a modest proteomic penetrance”. In contrast, they concluded that for future work, “The authors also recommend label-free approaches using data independent analysis (DIA), such as MS^E^ and all-ion-fragmentation, to enhance experimental repeatability”. All these observations suggest that this area is largely unexplored, but that, if it were investigated, it would result in an important contribution. 

Exosomes not only maintain a cytoplasmic environment for the glycosyltransferases to catalyze the surface glycan structure, but they also deliver extrinsic factors to the platelet surface, changing the platelets’ adhesive behavior. Platelet aggregation is also regulated by extrinsic factors such as exosomes from multiple cell types, which are altered by cigarette smoking acutely. Cell-derived extracellular vesicles are reported to contribute to platelet aggregation [93]. Interestingly, in our preliminary study, a short incubation of the nonsmokers’ platelets with the smokers’ exosomes increased their aggregation rates. The MS analysis of smokers’ and nonsmokers’ exosomes should be investigated to see if they show some differences in their nucleotide sugar contents, because the nucleotide sugars themselves are critical precursors to decorating the glycoprotein on the cell surface. Maybe such an alteration could favor the glycan-rich surface, specifically in xylose or mannose residues on the platelet. Maybe platelet-derived exosomes secreted by healthy volunteers inhibit platelet aggregation, while the cigarette smoking-mediated changes in the pool of the nucleotide sugars in exosomes enhance platelet activation. Additionally, the MS analysis of the proteins eluted from exosomes should be tested for similarities or differences between smoking and nonsmoking samples. Based on the source from which the exosomes are secreted, they have different contents, which play different roles in the circulatory system. Regardless, the contents of exosomes create an ideal unexploited reservoir for new biomarkers that could ultimately be adapted to point-of-care monitoring of disease-associated changes in circulation in real time.

These suggestions may provide new strategies for managing thrombotic risk induced by different diseases, such as stroke, diabetes, and cancer. This would overcome the most critical barrier to the field—the direct effects of antiplatelet treatments for those diseases —and greatly benefit the public. Future studies on the altering of platelets’ surface glycans would provide a novel pharmacological target, supporting a harm-reduction approach for treating and preventing platelet activation in different diseases. 

Enzymes can only catalyze their substrates in certain conditions, which are not found on the cell surface. However, the exosomes from multiple cell types shuttle the suitable conditions for the enzymatic reactions. This coordinates with the disease/disorder-associated supraphysiological level of 5-HT in blood plasma to remodel the platelet surface *N*-glycans. Indeed, in our preliminary findings, depending on the concentration of exosomes and the length of incubation time, platelets showed differences in aggregation rates; the exosomes isolated from physiological and supraphysiological levels of 5-HT-containing blood plasma represent the ultimate difference in making the platelets more prone to aggregation. This review summarized the remodeling of *N*-glycans on the platelet surface because of a coordination between exosomes and plasma free/unbound 5-HT at supraphysiological levels.

## 5. Conclusive Remarks

The stimulation of the sympathetic nervous system elevates 5-HT in blood plasma to supraphysiological levels. This pathological incident causes acute blood clots and contributes to cardiovascular disorders. This review brings various findings from different reports together to exhibit how plasma 5-HT at supraphysiological levels predisposes acute thrombus formation via changing the surface population and structures of *N*-glycans. Therefore, in the completion of the proposed studies in the fields would provide extensive information on the involvement of receptor blockers as novel pharmacological targets, supporting a harm-reduction approach for treating and preventing the blood clotting that are associated with elevated peripheral 5-HT levels due to various reasons, such as smoking, episodic hypertension, and cardiovascular disorders. 

## Figures and Tables

**Figure 1 ijms-25-11940-f001:**
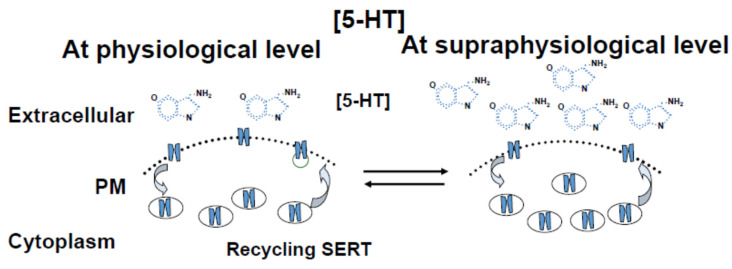
The density of SERT molecules on the plasma membrane exhibits a biphasic relationship to plasma 5-HT concentration ([5-HT]). An initial increase in [5-HT upregulates the density of SERT on platelet surface. However, a further increase in plasma [5-HT] downregulates the density of SERT molecules [13,46]. Indeed, our in vivo [13] and in vitro [46] studies confirm a dynamic relationship between extracellular 5-HT elevation, loss of surface SERT, and depletion of platelet 5-HT [45,48].

**Figure 2 ijms-25-11940-f002:**
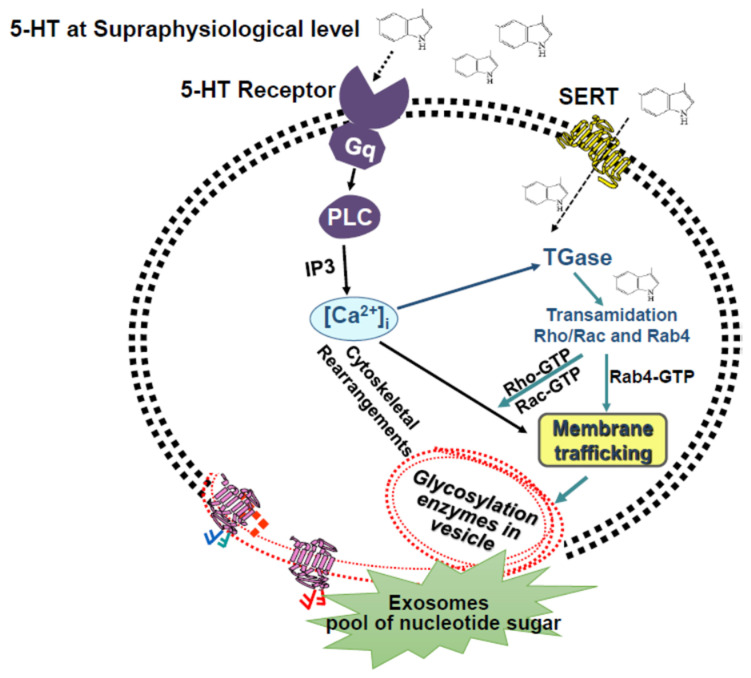
Proposed model for crosstalk between 5-HT receptors and SERT. In platelets, 5-HT signaling is mediated through 5-HT2A and G protein-coupled Gq receptors. These receptors transduce the signals via independent, but interconnected, pathways to promote platelet aggregation. The activation of 5-HT2A receptor signaling activates phospholipase C (PLC) and results in the hydrolysis of phosphatidylinositol 4,5-biphosphate (PIP2) to inositol-1,4,5-triphosphate (IP3) [17,25,27,28]. IP3 activates the serine/threonine protein kinase C and facilitates Ca^2+^ mobilization. Elevated free Ca^2+^ in the cytoplasm activates transglutaminase (TGase), which modifies Rab4/Rho/Rac with 5-HT (serotonylation) [9,25,27,28,29,30,31,44,45,46,48]. The Rab4 family of small GTPases regulates vesicular traffic [30,31,48]. I propose that in platelets, Rab4 is associated with early endosomes and regulates membrane recycling. In its active form (GTP bound), Rab4 facilitates the exocytosis of small vesicles that use the cytoskeletal network for their translocation to the plasma membrane. The translocation of small vesicles brings the glycosylation enzymes to the platelet surface.

**Table 1 ijms-25-11940-t001:** Proteins elevated on platelet surface upon smoking.

TABLE 1 LC/MS/MS Data: Proteins Elevated on Platelet Surface upon Smoking.			
Protein Names	Gene Names	Fold-Change	TTEST
UDP-glucuronic acid decarboxylase 1	UXS1	**Only on** **Smokers’ platelet**	0.000
Rho GTPase-activating protein	ARHGAP18	**Only on** **Smokers’ platelet**	0.044
18GDP-Man:Man(3)GlcNAc(2)-PP-Dol alpha-1,2-mannosyltransferase	ALG11	**13.4**	0.011
Rho-related GTP-binding protein RhoC	RHOC	9.16	0.026
Spectrin beta chain	SPTB	7.78	0.046
Tumor protein D54	TPD52L2	6.10	0.043
G-protein-signaling modulator 3	GPSM3	2.44	0.041
Zyxin	ZYX	2.26	0.044
Androgen-induced gene 1	AIG1	**Only on nonsmokers’ platelet**	0.001
Na/K-transporting ATPase subunit alpha-1	ATP1A1	**Only on nonsmokers’ platelet**	0.011

Data were processed with MaxQuant v1.5.3.1, and recalibrated MS and MS/MS data were searched against the combined forward and reverse Swiss-Prot human protein database [9]. Data were filtered with a 1% protein and peptide false discovery rate and required at least one unique peptide per protein. Protein intensity of detected peptides was calculated according to the area under the peak of extracted ions [9].

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
