# Peer review of "The Coordinated Changes in Platelet Glycan Patterns with Blood Serotonin and Exosomes"

_ijms, 2024, doi:10.3390/ijms252211940_

Round 1

Reviewer 1 Report

Comments and Suggestions for Authors

The article explores the relationship between blood serotonin (5-HT), platelet glycan patterns, and exosomes. It discusses how elevated levels of 5-HT influence platelet physiology, particularly through changes in glycan structures that affect platelet aggregation.

The topic is relevant given the increasing interest in the role of serotonin in cardiovascular health and pathology.

The article summarizes a wide range of literature, and the references list is comprehensive.

The integration of knowledge from biochemistry, cell biology, and cardiovascular physiology is commendable.

Areas to improve:

Please provide a deeper exploration of the biochemical pathways involved in glycan remodeling and exosome function. More detailed discussions on specific enzymes and their regulatory mechanisms would strengthen the argument.

Please elaborate more on the clinical implications of the findings. Discussing potential therapeutic targets or interventions based on the findings would enhance the article's impact.

There is lack of visual presentation in the review. For instance, a schematic representation of the interactions between 5-HT, exosomes, and platelet glycan changes could be beneficial.

Address the limitations of the current literature and areas of future improvement. 

Author Response

Thank you for your feedback and the questions you raised. My responses are as follows:

REVIEWER 1

  1. The topic is relevant;…the references list is comprehensive; the integration of

knowledge ….is commendable

Response: I appreciate the positive comments, thank you.

  1. If deeper exploration of the biochemical pathways in glycan remodeling and

exosome function…….. discussions on specific enzymes and their regulatory

mechanisms would strengthen the argument.

Response: The biochemical pathways and their descriptions are included in the revised manuscript as in two tables and in couple of paragraphs. Please see the

discussions on the specific enzymes and their regulatory mechanisms on page 6,

lines between 256-265.

  1. ..elaborate more on the clinical implications of the findings. Discussing potential

therapeutic targets or interventions based on the findings would enhance the

article's impact.

Response: Thank you. I worked on that and revised the article accordingly (page 8, Conclusion Remarks); furthermore, the following pages and lines cover this issue: Pg 2, lines 71-74, pg 5, lines 252-254, pg 6, lines 269-271, pg 7, lines 353, 354 and 361-364.

  1. …… visual presentation of the interactions between 5-HT, exosomes, and platelet glycan changes ….

Response: Excellent points and included two figures and a table in revised article.

  1. Compared …. the limitations of the current literature and areas of future

improvement.

Response: This is also covered in the revised article and in the remarks section.

I hope that this revision addresses the concerns raised by you, and I thank you for your valuable advice.

Reviewer 2 Report

Comments and Suggestions for Authors

- The background is not concise and clear. It is necessary to explain the biological function of serotonin in platelets and why is it important to understand the role of serotonin in platelets.

- The overall structure is confusing.

- In ”5-HTsignaling and platelets physiology", only SERT and 5-HT levels are mentioned. It would be necessary to include the downstream signaling pathway of 5-HT in platelets.

- "Discussion" section is missing.

- Recheck typo and grammars - frequent minor errors, e.g.:"Therefore, regulation of the SERT’s activity constitutes an important mech-anism in controlling the action of the 5-HT signaling becomes constitutively active when plasma 5-HT reaches a supraphysiological level." and " However, in systemic plasma measurement of 5-HT concentration appeared normal, inferring that this elevation could be locally elevated 5-HT levels may not be routinely detected as a risk factor.“

Author Response

Thank you for your feedback and the questions you raised. My responses are as follows:

REVIEWER 2

  1. …. … explain the biological function of serotonin in platelets …..why is it important to understand the role of serotonin in platelets.

Response: The background of 5-HT and its biological function are described in the first two pages starting from line 41-46.

  1. The overall structure is confusing.

Response: For giving a better understanding to the overall structure, two figures and a table are included in this revised version of the article.

  1. “5-HTsignaling and platelets physiology", only SERT and 5-HT levels are mentioned. It would be necessary to include the downstream signaling pathway of

5-HT in platelets.

Response: Thank you, the detail on the signaling pathways induced by 5-HT is presented in the new figures.

  1. "Discussion" section is missing.

Response: Each paragraph is closed through addressing the limitations in the field, as well as proposals to improve our understanding of mechanisms with some suggestions for a future improvement (Pg 2, lines 71-74, pg 5, lines 252-254, pg 6, lines 269-271, pg 7, lines 353, 354 and 361-364). Therefore, a separate discussion section is not included in the review article.

  1. Recheck typo and grammars - frequent minor errors

Response: I have edited and streamlined the entire text, paying close attention to

perfecting the presentation and clarifying the discussion.

I hope that this revision addresses the concerns raised by you, and I thank you for your valuable advice.